# Coexistence of Anaemia and Stunting among Children Aged 6–59 Months in Ethiopia: Findings from the Nationally Representative Cross-Sectional Study

**DOI:** 10.3390/ijerph20136251

**Published:** 2023-06-29

**Authors:** Biniyam Sahiledengle, Lillian Mwanri, Pammla Petrucka, Kingsley Emwinyore Agho

**Affiliations:** 1Department of Public Health, Madda Walabu University Goba Referral Hospital, Bale-Goba P.O. Box 302, Ethiopia; 2Research Centre for Public Health, Equity and Human Flourishing, Torrens University Australia, Adelaide Campus, Adelaide 5000, Australia; 3College of Nursing, University of Saskatchewan, Saskatoon, SK S7N 5A2, Canada; 4School of Health Sciences, Western Sydney University, Locked Bag 1797, Penrith 2751, Australia

**Keywords:** stunting, anaemia, concurrent stunting and anaemia, coexisting forms of malnutrition

## Abstract

Introduction: Stunting and anaemia, two severe public health problems, affect a significant number of children under the age of five. To date, the burden of and predictive factors for coexisting forms of stunting and anaemia in childhood have not been well documented in Ethiopia, where both the conditions are endemic. The primary aims of the present study were to: (i) determine the prevalence of co-morbid anaemia and stunting (CAS); (ii) and identify factors associated with these co-morbid conditions among children aged 6–59 months in Ethiopia. Methods: The study was based on data from the Ethiopian Demographic and Health Survey (EDHS 2005–2016). The EDHS was a cross-sectional study that used a two-stage stratified cluster sampling technique to select households. A total weighted sample of 21,172 children aged 6–59 months was included in the current study (EDHS-2005 (n = 3898), EDHS-2011 (n = 8943), and EDHS-2016 (n = 8332)). Children with height-for-age z-scores (HAZ) less than −2 SD were classified as stunted. Anaemia status was measured by haemoglobin level with readings below 11.0 g/deciliter (g/dL) categorized as anaemic. A multilevel mixed-effects logistic regression model was used to identify the factors associated with CAS. The findings from the models were reported as adjusted odds ratios (AOR) with 95% confidence intervals (CIs). Results: Almost half of the children were males (51.1%) and the majority were from rural areas (89.2%). The prevalence of CAS was 24.4% [95% CI: (23.8–24.9)]. Multivariate analyses revealed that children aged 12–23 months, 24–35 months, and 36–59 months, and children perceived by their mothers to be smaller than normal at birth had higher odds of CAS. The odds of CAS were significantly higher among children born to anaemic mothers [AOR: 1.25, 95% CI: (1.11–1.41)], mothers with very short stature [AOR: 2.04, 95% CI: (1.44–2.91)], children from households which practiced open defecation [AOR: 1.57, 95% CI: (1.27–1.92)], children born to mothers without education [AOR: 3.66, 95% CI: (1.85–7.22)], and those who reside in rural areas [AOR: 1.41, 95% CI: (1.10, 1.82)]. Male children had 19% lower odds of having CAS compared to female children [AOR: 0.81, 95% CI: (0.73–0.91)]. Children born to mothers who had normal body mass index (BMI) [AOR: 0.82, 95%CI: (0.73–0.92)] reported lower odds of CAS. Conclusions: One in four preschool-age children in Ethiopia had co-morbid anaemia and stunting, which is a significant public health problem. Future interventions to reduce CAS in Ethiopia should target those children perceived to be small at birth, anaemic mothers, and mothers with short stature.

## 1. Introduction

Undernutrition is one of the significant public health scourges that particularly affects young children [1], and around 45% of deaths among children under 5 years of age are linked to undernutrition [2]. In 2019, 155 million children globally under the age of 5 years were stunted [3]. During the same period, the prevalence of anaemia in children aged 6–59 months was 39.8%, equivalent to 269 million children. Africa was the hardest hit region, with anaemia affecting 60.2% of children aged 6–59 months [4], and 35% of children under the age of five were stunted [5].

Stunting and anaemia may occur at the same time and have negative long-term health consequences [6,7]. Although it is well-recognised that children can suffer from multiple forms of malnutrition, this syndemic remains an understudied phenomenon on a global scale [8]. A few previous studies have attempted to estimate the coexistence of anaemia and stunting (CAS). For instance, the prevalence of CAS in forty-three countries in LMICs was estimated to be 21.5%, ranging from 2.6% to 46.1% [6], 5.9% in Venezuela [9], 21.5% in India, and 30.4% in Peru [7]. In Ethiopia, evidence regarding CAS among children under five years old is limited and few studies have conveyed the co-morbidities in children 6–59 months, with the prevalence ranging from 17.8 to 23.9% [10,11].

Children who suffer from multiple forms of malnutrition such as CAS are more likely to have poor health and experience significant health problems, including impaired cognitive and developmental consequences [6,9,12]. Some studies have attempted to identify the common factors linked to anaemia and child growth retardation [7,13]. Both stunting and anaemia have often been linked to several chronic diseases, and they might share overlapping risk factors [14]. The contributing factors to concurrent forms of malnutrition are multifold including the child’s age [10,11], a lack of iron supplementation during pregnancy [10], living in a severely food insecure household [10], rural residence [6,11], low household wealth [6,11], low educational level of caregivers [6,11], male sex [11], history of infections [11], and a small birth size [11].

According to the recent 2016 Ethiopian Demographic and Health Survey (EDHS), the prevalence of anaemia among children aged 6–59 months was reported to be 57%. At the same time, 37% of children under the age of five were reported as stunted [15]. The World Health Organization (WHO) classifies both conditions as major public health issues [16,17].

To address undernutrition and micronutrient deficiency among Ethiopian children, Ethiopia has developed several strategies, including the National Nutrition Strategy and the National Nutrition Programmes (NNP) [18,19]. However, the prevalence of anaemia and stunting is still alarmingly high [20,21]. Furthermore, the concurrent forms of these two conditions, anaemia and stunting, are a relatively new issue that has yet to be fully explored in Ethiopia. 

As far as we are aware, studies that capture the prevalence and risk factors for coexisting stunting and anaemia were limited in Ethiopia. Prior studies by Orsango and colleagues [10], Amare and Lindtjorn [22], and Mohammed and colleagues were based on the analysis of EDHS data that focused only on children aged 6–23 months [11]. To our knowledge, there are no population-level studies that have examined CAS using large pooled data in Ethiopia. Therefore, the present study aimed to: (i) determine the prevalence of CAS (i.e., coexisting stunting and anaemia) among children aged 6–59 months in Ethiopia, and (ii) identify the factors associated with these co-morbid conditions. Understanding the burden of concurrent forms of malnutrition and triggering factors is essential to developing effective policies and allocating sufficient resources to enact them.

## 2. Materials and Methods

### 2.1. Setting, Data Source, and Study Design 

The study used data from the Ethiopian Demographic and Health Survey (EDHS) from 2005 to 2016 [23,24,25]. The EDHS was a cross-sectional study that employed a two-stage stratified cluster sampling technique to select households. The sampling procedure has been described in detail in the EDHS report [23,24,25]. For this study, 21,172 children’s data from the EDHS (2005–2016) documents were obtained and used for this secondary analysis, with a complete response to all variables of interest (Appendix A). 

### 2.2. Data Collection

The EDHS gathered information on children’s nutritional status by measuring the weight and height of children under the age of five in all sampled households. Height was measured with a measuring board (i.e., ShorrBoard Portable height length measurement board). Children younger than the age of 24 months were measured lying down on the board (recumbent length) while standing height was measured for older children [23]. The EDHS also collected blood samples from all children aged 6 to 59 months who participated in the survey for haemoglobin tests. Haemoglobin analysis was carried out on-site using a battery-operated portable HemoCue analyzer, Angelholm, Sweden (*HemoCue^®^*). Blood samples were drawn from a drop of blood taken from a finger prick. Children with a haemoglobin level of less than 11 g/dL were considered anaemic [23,24,25]. 

The EDHS collected anthropometric data on height and weight for women age 15–49 who were not pregnant. These data were used to calculate maternal body mass index (BMI). The BMI was calculated by dividing weight in kilograms by height in metres squared (kg/m^2^). Maternal BMI was classified as underweight (<18.5 kg/m^2^), normal (18.5 to <24.9 kg/m^2^), or overweight/obesity ≥ 25.0 kg/m^2^) [23]. 

Anaemia among women age 15–49 was measured by the HemoCue instrument capillary blood, and was collected exclusively from a finger prick. A hemoglobin level of less than 11 g/dL was categorized as anaemia for non-pregnant women [23,26]. 

### 2.3. Variable and Measurements

Concurrent stunting and anaemia (CAS) was our primary outcome, defined as the child having a simultaneous presence of both anaemia and stunting conditions [10,11]. All children with height-for-age z-scores (HAZ) less than −2 standard deviations (SD) were classified as stunted [27] and a haemoglobin level < 11.0 g/deciliter was categorized as anaemia [26]. 

### 2.4. Independent Variable

Potential factors of CAS in children were extracted from the EDHS dataset. The factors were also selected based on previous studies [6,7,10,11]. The identified factors were categorized into individual, household, and community-level factors. A detailed list and variable coding of all independent variables are presented in Appendix A.

### 2.5. Data Analysis 

All analyses were carried out using STATA/MP version 14.1 (StataCorp, College Station, TX, USA) to adjust for clusters and survey weights. Sampling weighting was applied to all descriptive statistics to compensate for the disproportionate allocation of the sample [23]. Multilevel logistic regression models were used to determine community- and individual-level factors associated with CAS. A bivariable multilevel analysis was first performed to identify factors associated with CAS. Potential factors with a *p*-value < 0.2 obtained in the multilevel bivariable analysis were selected to enter multilevel multivariable logistic regression models to estimate their independent association with the outcome variable. Four models were fitted and an empty model without any explanatory variables was run to detect the presence of a possible contextual effect; the first with individual-level variables (*model I*), the second with household-level variables (*model II*), the third with community-level variables (*model III*), and the fourth with individual-, household-, and community-level variables (*model IV*). The intraclass correlation coefficient (ICC) was computed for each model to show the variations explained at each level of modelling. Model comparisons were performed using the deviance information criteria (DIC), Akaike information criteria (AIC), and Bayesian information criteria (BIC). The model with the lowest DIC, AIC, and BIC, was considered the best-fit model. Finally, the fourth model (*model IV*) with the lowest information criteria value was chosen as the final best-fit model. 

## 3. Results 

### 3.1. Sociodemographic Characteristics of the Sample

A total of 21,172 children aged 6–59 months were included in the study (EDHS-2005 (*n* = 3898), EDHS-2011 (*n* = 8943), and EDHS-2016 (*n* = 8332)) and the majority were from rural areas (89.2%). Almost half of the children were males (51.1%). Nearly half of the children (45.9%) were in the age group of 36–59 months and 21% were between the ages of 12–23 and 14–35 months. The majority of the children (70.5%) were currently breastfeeding. Of all the study participants, 14,862 (70.2%) were born to mothers who were not educated, 45.5% were from poor households, 21.0% were from middle wealth quantiles, and 33.5% were from the richest households (Table 1).

### 3.2. Prevalence of Anaemia, Stunting, and Concurrent Anaemia and Stunting (CAS)

The prevalence of stunting and anaemia was found to be 43.1% (95%CI: 42.4–43.7) and 49.3% (95%CI: 48.7–49.9), respectively. The prevalence of CAS was 24.4% [95%CI: (23.8–24.9)]. Between 2005 and 2011, the prevalence of CAS among Ethiopian children declined from 26.6% (95%CI: 25.3–28.1) to 22.4% (95%CI: 21.5–23.3), but increased to 25.5% (95%CI: 24.5–26.4) in 2016 (Figure 1).

The prevalence of CAS varied by gender, with males accounting for 54.0% of those affected. About 41.9% of CAS children were between 36 and 59 months. The proportion of children with CAS was higher among uneducated mothers (75.1%), and in mothers who did not attend ANC (57.1%). The children from the poorest households (52.9%) and the children living in rural regions had a higher prevalence of CAS (93.6%). The detailed prevalence by different factors is given in Table 2.

### 3.3. Factors Associated with Concurrent Anaemia and Stunting (CAS)

Table 2 also displays the results of a multilevel bivariate analysis of CAS. Table 3 shows the covariate-adjusted estimates based on the hierarchical logistic regression analyses. The odds of experiencing CAS were lower among male children in comparison to females [AOR: 0.81, 95% CI: (0.73–0.91)]. The odds of CAS were 3.07 times more likely [AOR: 3.07, 95% CI: (2.58–3.66)] for children aged 12–23 months, 4.13 times more likely [AOR: 4.13, 95% CI: (3.40–5.01)] for children aged 24–35 months, and 2.51 times more likely [AOR: 2.51, 95% CI: (1.97–3.20)] for children aged 36–59 months. The odds of having CAS were lower among children born from mothers with a BMI (kg/m^2^) of 18.5 to 24.9 (normal) [AOR: 0.82, 95% CI: (0.73–0.92)], and 25 and above (overweight or obese) [AOR: 0.60, 95% CI: (0.45–0.81)] as compared to those born from underweight mothers. The odds of CAS were significantly higher in children born to anaemic mothers [AOR: 1.25, 95% CI: (1.11–1.41)], and mothers with very short stature [AOR: 2.04, 95% CI: (1.44–2.91)] and short stature [AOR: 1.48, 95% CI: (1.32–1.65)]. Those children perceived by their mother to be smaller than normal at birth [AOR: 1.47, 95% CI: (1.28–1.69)], and children from households with open defecation [AOR: 1.57, 95% CI: (1.27–1.92)] reported higher odds of CAS. The odds of CAS among children born to mothers having no education [AOR: 3.66, 95% CI: (1.85–7.22)], those with primary education [AOR: 3.48, 95% CI: (1.77, 6.84)], and secondary education [AOR: 3.13, 95% CI: (1.54, 6.37)] were higher compared to children of mothers who had achieved higher-level education.

The odds of having CAS among children from households that cooked outdoors [AOR: 0.82, 95% CI: (0.69–0.97)] was lower than children from households that cooked in the house. At the community level, the odds of CAS were higher among those residing in rural areas [AOR: 1.41, 95% CI: (1.10, 1.82)] than urban dwellers. The odds of having CAS among children from large central regions were lower compared with children in the metropolis area [AOR: 0.79, 95% CI: (0.63–0.98)] (Table 3).

## 4. Discussion

The present study was mainly designed to elucidate the prevalence of co-morbid anaemia and stunting (CAS), and its associated factors, among children aged 6–59 months. To our knowledge, no study in Ethiopia has examined the coexisting forms of CAS using a country-wide pooled dataset. In Ethiopia, the prevalence of CAS was found to be 24.40%. The factors associated with increased odds of CAS include older child age, children reported to be smaller than normal at birth, children born from anaemic mothers, children born to mothers with short or very short stature, children born to mothers with normal weight or overweight/obese, and children born to mothers with no education. In addition, higher odds of CAS were reported among children from households that practiced open defecation and those living in rural areas.

This study reported a 24.4% prevalence of CAS, which was comparable with a previous study conducted in Ethiopia that reported a 23.9% prevalence [11]. This finding implies that CAS in Ethiopia is a major public health concern, as it affects one in every four children under the age of five years. Several explanations are plausible for the higher prevalence of CAS in under-five-year-olds in Ethiopia. Firstly, both stunting and anaemia have been reported to be the most prevalent conditions in Ethiopia. The national prevalence of stunting was 38% and more than half (57%) of Ethiopian children aged 6–59 months are anaemic [23]. Secondly, Ethiopian poor dietary patterns and poor food security may explain the high prevalence of concurrent undernutrition and micronutrient deficiencies. According to an Ethiopian study, food-insecure households may have limited access to a diverse range of nutritious foods and have low meal frequencies, which increase the co-occurrence of stunting and anaemia due to overall low nutrient intakes [11].

The finding on the current prevalence of CAS was slightly higher than the 21.5% reported in a study conducted in forty-three LIMICs among younger children aged 6 to 59 months [6], and the observed co-occurrence of 22% among Indian children aged 6–18 months [7]. Additionally, the prevalence found in this study was higher than that reported in a community-based cross-sectional survey conducted in Southern Ethiopia, where 17.8% [10] and 10.5% had CAS [22], and than the 5.94% reported in Maracaibo, Venezuela [9]. This variation could be attributed to the study design and sample size, as we used national data sources.

In this study, the odds of experiencing CAS were lower among male children in comparison to females. However, other studies have reported higher odds of CAS in boys than girls [11,28]. This could be explained, in part, by the difference in the study population, as the previous two studies included children aged 6–24 months in their analysis. The role of sex varies across places, and its relationship with the burden of CAS requires further investigation.

Our finding reported higher odds of CAS in older children than in their younger counterparts which concurred with previous reports, which consistently demonstrated higher risks of CAS as the children’s age increased [10,11]. The data from the current study confirmed the lower odds of comorbid anaemia and stunting in children whose mothers have normal BMI than those born to underweight mothers. This finding indicates that children born to mothers with normal BMIs are less likely to experience anaemia, stunting, or CAS. A prior prospective cohort study in China on maternal BMI during early pregnancy with infant anaemia showed that maternal BMI during early pregnancy is correlated with infant haemoglobin in an inverse U-shaped profile [29]. Underweight mothers are more likely to have a stunted child, and stunted children are more likely to be anaemic, due to the intergenerational cycle of stunting. Supporting this assertion, a Brazilian study found positive correlations between haemoglobin (Hb) levels and HAZ, even after controlling for age [30], which ultimately led to CAS. Furthermore, Afolabi and Palamuleni reported that the likelihood of stunting was higher among anaemic children [31]. Likewise, the study identified children born of anaemic mothers and mothers with short/very short stature as being significantly associated with higher odds of childhood CAS. Various earlier studies have reported that the odds of stunting are higher among children of mothers with short stature [32,33,34], which may be a plausible reason for higher CAS. Moreover, the reported intergenerational cycle of malnutrition, in which stunted female children grow to become stunted mothers, who give birth to stunted children, may explain the relationship [35]. The current study also found that children who were reported to be smaller than normal at birth, and those from households with open defecation, had a higher risk of CAS. These findings were consistent with the existing literature, demonstrating that a small birth size is associated with a higher risk of CAS than a large birth size [11]. In the current study, children from households practicing open defecation had increased odds of having CAS. The association between anaemia and/or stunting and open defecation could have many plausible explanations, including being caused by intestinal parasite infections and environmental enteropathy [36,37,38,39,40].

The odds of having CAS among children from households that cooked outdoors was lower than children from households that cooked in the house. We hypothesis that, in a country where a significant number of households used solid fuel and no in-house potable water, cooking outdoors could pose a lower risk of indoor air pollution and indoor contaminations due to an enclosed space and a lack of running water. Other studies have demonstrated that the likelihood of anaemia was higher among children from households that used solid fuels; biofuel smoke contains significant amounts of carbon monoxide [41,42].

In the current study, the education level of the mother was also associated with CAS. For example, mothers with no education, and those with primary and secondary education were associated with higher odds of CAS compared with those with higher than secondary level education. This finding is also consistent with previous studies that found no or low maternal education status to be associated with several poor nutritional outcomes [6,7,11]. Additionally, among the current study findings, children living in rural areas had higher odds of having CAS than those living in urban areas. This finding aligns with previous double-burden studies that showed living in rural areas to be strongly associated with co-morbidities [6,43]. People in rural areas are often at a disadvantage in terms of living conditions, economic status, and access to health care and other social services across many LMICs.

The study has some limitations. Firstly, the analyses were conducted using EDHS data collected in a cross-sectional survey, which prevents causal inferences. Secondly, because of the self-reported nature of the inquiry, there is a possibility of recall bias. Thirdly, due to the secondary nature of the data, the present study was limited by unmeasured confounders.

## 5. Conclusions

One in every four children in Ethiopia is affected by CAS, making it a significant public health problem. Stunting, anaemia, and their co-occurring health conditions have created a syndemic situation in Ethiopia situation whereby two or more factors are likely to work together to exacerbate a health crisis. Both individual-level (older children, children perceived by their mother to be smaller than normal at birth, children born to anaemic mothers, mothers with short or very short stature, low maternal education, and children from households with open defecation) and community-level factors (place of residency) were revealed to be important factors for CAS in Ethiopia. While the prevalence of CAS was lower among male children, children born from mothers with BMIs (kg/m^2^) of 18.5 to 24.9 and 25 and above, children born by mothers with BMIs lower that 18.5 kg/m^2^, and children born from mothers who were anaemic were likely to have CAS. Thus, intervention programs aiming to prevent anaemia and stunting should also consider addressing both burdens simultaneously by identifying individual- and contextual-level factors.

## Figures and Tables

**Figure 1 ijerph-20-06251-f001:**
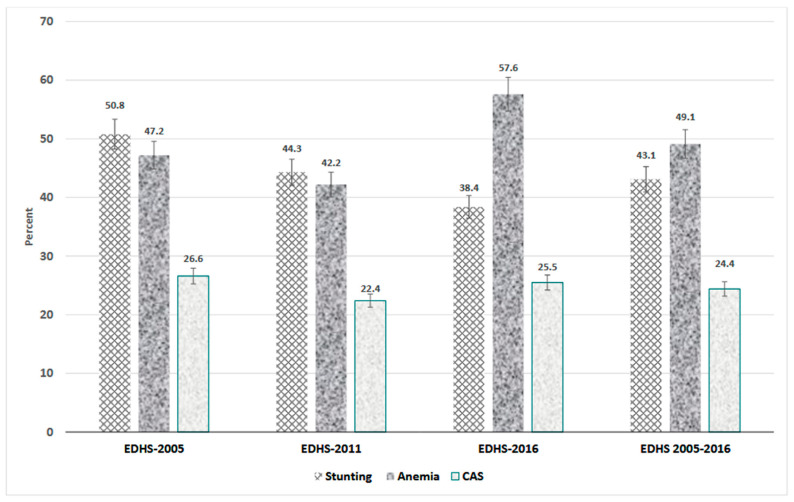
The prevalence of stunting, anaemia, and CAS among children 6–59 months in Ethiopia, EDHS (2005–2016).

**Table 1 ijerph-20-06251-t001:** Distributions of individual-, household-, and community-level characteristics, EDHS (2005–2016).

Variables	Total Weighted Frequency	Percent
*Individual-level factors*		
*Child factors*		
Sex		
Male	10,824	51.1
Female	10,348	48.9
Age (months)		
6–11	2479	11.7
12–23	4504	21.3
24–35	4471	21.1
36–59	9719	45.9
Birth order		
Firstborn	3698	17.5
2–4	9206	43.5
5 or higher	8267	39.0
Birth interval		
<33 months	14,566	68.8
≥33 months	6606	31.2
Perceived size of a child at birth		
Larger	6792	32.2
Average	8574	40.6
Small	5739	27.2
Currently breastfeeding		
Yes	14,936	70.5
No	6236	29.5
Received measles		
Yes	8086	47.5
No	8925	52.5
Full vaccination		
Yes	4156	28.9
No	12,546	75.1
Diarrhea		
Yes	3065	14.5
No	18,077	85.5
Fever		
Yes	3564	16.9
No	17,569	83.1
Children age 6–59 months given iron supplement		
Yes	1336	6.3
No	19,836	93.7
Children age 6–59 months given deworming medication		
Yes	2982	14.1
No	18,190	85.9
*Parental factors*		
Mother’s age		
<18	108	0.5
18–24	4524	21.4
25–34	11,146	52.6
≥35	5394	25.5
Mother’s education		
No education	14,862	70.2
Primary	5321	25.1
Secondary	678	3.2
Higher	311	1.5
Mother’s occupation		
Not working	11,184	53.0
Non agriculture	4659	22.1
Agriculture	5239	24.8
ANC Visit		
None	7081	51.4
1–3	3438	24.9
4–7	2929	21.3
8+	318	2.3
Maternal BMI (kg/m^2^)		
<18.5	4416	21.0
18.5 to 24.9	15,609	74.2
25+	1001	4.8
Any anaemia		
Yes	5070	24.5
No	15,601	75.5
Maternal stature		
Very short (<145 cm)	466	2.2
Short (145 to <155 cm)	7337	34.9
Normal/Tall (155 to <200 cm)	13,237	62.9
Listening to radio		
Not at all	13,182	62.3
Yes	7983	37.7
Watching television		
Not at all	16,532	78.1
Yes	4625	21.9
*Household factors*		
Wealth index		
Poor	9640	45.5
Middle	4439	21.0
Rich	7094	33.5
Household Size		
1–4	5001	23.6
≥5	16,171	76.4
Place of cooking		
In the house	8586	49.7
In separate building	7089	41.1
Outdoors	1585	9.2
Type of cooking fuel		
Clean fuels	349	1.7
Solid fuels	20,441	98.3
Toilet facility		
Improved	2127	10.2
Unimproved	9368	44.8
Open defecation	9421	45.0
Source of drinking water		
Improved	9571	45.8
Unimproved	11,338	54.2
Household flooring		
Improved	1892	8.9
Unimproved	19,274	91.1
Time to reach a water source		
On premises	1489	7.0
≤30 min	11,863	56.0
31–60 min	4520	21.4
>60 min	3300	15.6
*Community Level Factors*		
Residence		
Urban	2279	10.8
Rural	18,894	89.2
Region ^#^		
Large centrals	19,459	91.9
Small peripherals	1220	5.8
Metropolis	493	2.3
Ecological Zone		
Tropical zone	3148	14.9
Subtropical zone	14,924	70.5
Cool zone	3100	14.6
EDHS		
2005	3898	18.4
2011	8943	42.2
2016	8332	39.4

CAS: coexistence of anaemia and stunting; maternal BMI: thin (<8.5 kg/m^2^), normal (18.5–24.9 kg/m^2^), and overweight or obese (>=25.0 kg/m^2^). ^#^: The geographical region of Ethiopia where household heads live. Tigray, Amhara, Oromia, and Southern Nations Nationalities and Peoples’ Region (SNNPRs) were categorized under Large central regions; Afar, Somali, Benishangul, and Gambella were under Small peripherals, while Metropolis included Harari, Dire Dawa, and Addis Ababa regions.

**Table 2 ijerph-20-06251-t002:** Bivariable multilevel logistic regression analysis of factors associated with co-morbid anaemia and stunting among children in 6–59 months Ethiopia, EDHS (2005–2016).

Variables	Concurrent Anaemia and Stunting (CAS), n (%)		Crude OR	*p*-Value
	Yes	No		
*Child factors*				
Sex				
Male	2789 (54.0)	8035 (50.2)	1.14 (1.07–1.22)	*p* < 0.001
Female	2376 (46.0)	7972 (49.8)	Ref.	
Age (months)				
6–11	380 (7.4)	2098 (13.1)	Ref.	
12–23	1326 (25.7)	3178 (19.9)	2.69 (2.34–3.09)	*p* < 0.001
24–35	1297 (25.1)	3173 (19.8)	2.95 (2.56–3.39)	*p* < 0.001
36–59	2163 (41.9)	7556 (47.2)	1.89 (1.65–2.16)	*p* < 0.001
Birth order				
First born	812 (15.7)	2886 (18.0)	Ref.	
2–4	2186 (42.3)	7020 (43.9)	1.18 (1.07–1.29)	0.001
5 or higher	2167 (41.9)	6100 (38.1)	1.40 (1.27–1.55)	*p* < 0.001
Birth interval				
<33 months	3569 (69.1)	10,996 (68.7)	0.98 (0.92–1.06)	0.767
≥33 months	1596 (30.9)	5010 (31.3)	Ref.	
Perceived size of a child at birth				
Larger	1555 (30.2)	5.236 (32.8)	Ref.	
Average	1969 (38.2)	6604 (41.4)	1.09 (1.01–1.19)	0.03
Small	1627 (31.6)	4112 (25.8)	1.45 (1.33–1.58)	*p* < 0.001
Currently breastfeeding				
Yes	3809 (73.7)	11,127 (69.5)	1.08 (1.01–1.16)	0.022
No	1356 (26.3)	4879 (30.5)	Ref.	
Full vaccination				
Yes	941 (23.0)	3215 (25.5)	Ref.	
No	3153 (77.0)	9393 (74.5)	1.21 (1.11–1.32)	*p* < 0.001
Diarrhea				
Yes	841 (16.3)	2224 (13.9)	1.25 (1.14–1.37)	*p* < 0.001
No	4319 (83.7)	13,758 (86.1)	Ref.	
Fever				
Yes	959 (18.6)	2605 (16.3)	1.19 (1.09–1.29)	*p* < 0.001
No	4196 (81.4)	13,373 (83.7)	Ref.	
Children age 6–59 months given iron supplement				
Yes	309 (6.0)	1026 (6.4)	Ref.	
No	4856 (94.0)	14,980 (93.6)	1.03 (0.91–1.18)	0.623
Children age 6–59 months given deworming medication				
Yes	652 (12.6)	2330 (14.6)	Ref.	
No	4513 (87.4)	13,677 (85.4)	1.18 (1.07–1.31)	0.001
*Parental factors*				
Mother’s age				
<18	33 (0.6)	75 (0.5)	0.90 (0.56–1.44)	0.665
18–24	1110 (21.5)	3413 (21.3)	0.96 (0.87–1.06)	0.490
25–34	2727 (52.8)	8419 (52.6)	0.99 (0.92–1.08)	0.981
≥35	1295 (25.1)	4099 (25.6)	Ref.	
Mother’s education				
No education	3881 (75.1)	10,980 (68.6)	6.06 (3.95–9.29)	*p* < 0.001
Primary	1159 (22.4)	4.162 (26.0)	4.35 (2.83–6.69)	*p* < 0.001
Secondary	102 (2.0)	576 (3.6)	2.65 (1.67–4.22)	*p* < 0.001
Higher	23 (0.5)	287 (1.8)	Ref.	
Mother’s occupation				
Not working	2828 (54.9)	8355 (52.4)	1.29 (1.18–1.41)	*p* < 0.001
Agriculture	1312 (25.5)	3927 (24.6)	1.24 (1.11–1.39)	*p* < 0.001
Non agriculture	1004 (19.5)	3656 (22.9)	Ref.	
ANC visit				
None	1949 (57.1)	5132 (49.6)	2.11 (1.63–2.72)	*p* < 0.001
1–3	817 (23.9)	2621 (25.3)	1.71 (1.32–2.22)	*p* < 0.001
4–7	591 (17.3)	2337 (22.6)	1.18 (0.91–1.54)	0.219
8+	54 (1.6)	264 (2.5)	Ref.	
Maternal BMI (kg/m^2^)				
<18.5	1215 (23.7)	3202 (20.1)	Ref.	
18.5 to 24.9	3761 (73.4)	11,848 (74.5)	0.80 (0.74–0.87)	*p* < 0.001
25+	149 (2.9)	852 (5.4)	0.43 (0.36–0.51)	*p* < 0.001
Any anaemia				
Yes	1550 (30.4)	3520 (22.6)	1.42 (1.32–1.53)	*p* < 0.001
No	3543 (69.6)	12,058 (77.4)	Ref.	
Maternal stature				
Very short (<145 cm)	157 (3.1)	309 (1.9)	1.77 (1.41–2.22)	*p* < 0.001
Short (145 to <155 cm)	2059 (40.2)	5278 (33.2)	1.39 (1.29–1.49)	*p* < 0.001
Normal/Tall (155 to <200 cm)	2909 (56.7)	10,328 (64.9)	Ref.	
Listening to radio				
Not at all	3332 (64.6)	9849 (61.5)	1.25 (1.17–1.35)	*p* < 0.001
Yes	1828 (35.4)	6155 (38.5)	Ref.	
Watching television				
Not at all	4197 (81.3)	12,335 (77.1)	1.71 (1.56–1.87)	*p* < 0.001
Yes	968 (18.7)	3657 (22.9)	Ref.	
*Household factors*				
Wealth index				
Poor	2732 (52.9)	6908 (43.2)	1.90 (1.75–2.06)	*p* < 0.001
Middle	1048 (20.3)	3391 (21.2)	1.39 (1.25–1.55)	*p* < 0.001
Rich	1386 (26.8)	5707 (35.6)	Ref.	
Household size				
1–4	1114 (21.6)	3888 (24.3)	0.83 (0.77–0.90)	*p* < 0.001
≥5	4052 (78.4)	12,119 (75.7)	Ref.	
Place of cooking				
In the house	2324 (56.3)	6262 (47.7)	Ref.	
In separate building	1438 (34.9)	5651 (43.0)	0.72 (0.66–0.79)	*p* < 0.001
Outdoors	362 (8.8)	1223 (9.3)	0.79 (0.71–0.89)	*p* < 0.001
Type of cooking fuel				
Clean fuels	41 (0.8)	308 (2.0)	Ref.	
Solid fuels	5019 (99.2)	15,421 (98.0)	3.69 (2.72–5.01)	*p* < 0.001
Toilet facility				
Improved	352 (6.9)	1774 (11.2)	Ref.	
Unimproved	2147 (42.1)	7221 (45.7)	1.57 (1.39–1.77)	*p* < 0.001
Open defecation	2599 (51.0)	6822 (43.1)	2.18 (1.94–2.44)	*p* < 0.001
Source of drinking water				
Improved	2228 (43.7)	7343 (46.4)	Ref.	
Unimproved	2870 (56.3)	8468 (53.6)	1.24 (1.15–1.33)	*p* < 0.001
Household flooring				
Improved	251 (4.9)	1641 (10.3)	Ref.	
Unimproved	4915 (95.1)	14,359 (89.7)	2.28 (2.02–2.57)	*p* < 0.001
Time to reach a water source				
On premises	199 (3.9)	1289 (8.1)	Ref.	
≤30 min	2889 (55.9)	8973 (56.1)	1.94 (1.70–2.23)	*p* < 0.001
31–60 min	1197 (23.2)	3322 (20.7)	2.26 (1.95–2.63)	*p* < 0.001
>60 min	879 (17.0)	2421 (15.1)	2.46 (2.12–2.85)	*p* < 0.001
*Community Level Factors*				
Residence				
Urban	333 (6.4)	1946 (12.2)	Ref.	
Rural	4833 (93.6)	14,060 (87.8)	2.22 (1.98–2.48)	*p* < 0.001
Region				
Large centrals	4763 (92.2)	14,696 (91.8)	1.22 (1.09–1.37)	0.001
Small peripherals	335 (6.5)	885 (5.5)	1.49 (1.32–1.69)	*p* < 0.001
Metropolis	67 (1.3)	426 (2.7)	Ref.	
Ecological zone				
Tropical zone	741 (14.3)	2406 (15.0)	0.92 (0.80–1.04)	0.199
Subtropical zone	3532 (68.4)	11,392 (71.2)	0.71 (0.62–0.81)	*p* < 0.001
Cool zone	892 (17.3)	2208 (13.8)	Ref.	
EDHS				
2005	1038 (20.1)	2859 (17.9)	1.01 (0.91–1.10)	0.954
2011	2004 (38.8)	6939 (43.3)	0.92 (0.85–0.99)	0.029
2016	2124 (41.1)	6208 (38.8)	Ref.	

**Table 3 ijerph-20-06251-t003:** Multivariable multilevel logistic regression analysis to determine associated factors of concurrent anaemia and stunting (CAS) among children 6–59 months in Ethiopia, EDHS (2005–2016).

Variables	Null Model	Model I ^a^	Model II ^b^	Model III ^c^	Model IV ^d^
		AOR (95%CI)	AOR (95%CI)	AOR (95%CI)	AOR (95%CI)
*Individual-Level Factors*					
*Child factors*					
Sex					
Male		0.80 (0.73–0.88) **			0.81 (0.73–0.91) **
Female		Ref.			Ref.
Age (months)					
6–11		Ref.			Ref.
12–23		2.83 (2.43–3.28) **			3.07 (2.58–3.66) **
24–35		3.55 (3.01–4.18) **			4.13 (3.40–5.01) **
36–59		2.17 (1.78–2.64) **			2.51 (1.97–3.20) **
Birth order					
First born		Ref.			Ref.
2–4		1.08 (0.94–1.24)			1.01 (0.85–1.17)
5 or higher		1.18 (1.03–1.37) *			1.09 (0.90–1.32)
Perceived size of a child at birth					
Larger		Ref.			Ref.
Average		1.12 (1.01–1.26) *			1.09 (0.95–1.25)
Small		1.47 (1.30–1.66) **			1.47 (1.28–1.69) **
Currently breastfeeding					
Yes		1.13 (1.01–1.27) *			1.11 (0.97–1.27)
No		Ref.			Ref.
Full vaccination					
Yes		Ref.			Ref.
No		1.08 (0.97–1.21)			1.02 (0.89–1.16)
Diarrhea					
Yes		1.11 (0.98–1.25)			1.09 (0.94–1.25)
No		Ref.			Ref.
Fever					
Yes		1.07 (0.95–1.19)			1.11 (0.97–1.27)
No		Ref.			Ref.
Children age 6–59 months given deworming medication					
Yes		Ref.			Ref.
No		1.13 (0.98–1.30)			1.11 (0.95–1.28)
*Parental factors*					
Mother’s education					
No education		4.13 (2.25–7.57) **			3.66 (1.85–7.22) **
Primary		3.64 (1.99–6.65) **			3.48 (1.77–6.84) **
Secondary		2.95 (1.56–5.56) *			3.13 (1.54–6.37) *
Higher		Ref.			Ref.
Mother’s occupation					
Not working		1.15 (1.01–1.29) *			1.03 (0.90–1.19)
Agriculture		1.11 (0.96–1.28)			0.98 (0.83–1.16)
Non agriculture		Ref.			Ref.
ANC visit					
None		1.14 (0.83–1.55)			0.86 (0.60–1.24)
1–3		1.05 (0.77–1.44)			0.81 (0.56–1.17)
4–7		0.84 (0.62–1.14)			0.71 (0.49–1.01)
8+		Ref.			Ref.
Maternal BMI (kg/m^2^)					
<18.5		Ref.			Ref.
18.5 to 24.9		0.81 (0.73–0.89) **			0.82 (0.73–0.92) *
25+		0.53 (0.41–0.69) **			0.60 (0.45–0.81) *
Any anaemia					
Yes		1.30 (1.17–1.44) **			1.25 (1.11–1.41) **
No		Ref.			Ref.
Maternal stature					
Very short (<145 cm)		1.93 (1.41–2.62) **			2.04 (1.44–2.91) **
Short (145 to <155 cm)		1.43 (1.29–1.57) **			1.48 (1.32–1.65) **
Normal/Tall (155 to <200 cm)		Ref.			Ref.
Listening to radio					
Not at all		0.96 (0.86–1.07)			0.88 (0.78–1.01)
Yes		Ref.			Ref.
Watching television					
Not at all		1.33 (1.16–1.52) **			1.07 (0.91–1.25)
Yes		Ref.			Ref.
*Household factors*					
Wealth index					
Poor			1.38 (1.22–1.57) **		1.15 (0.95–1.37)
Middle			1.09 (0.95–1.25)		0.98 (0.81–1.18)
Rich			Ref.		Ref.
Household size					
1–4			0.88 (0.81–0.97) *		0.88 (0.77–1.02)
≥5			Ref.		Ref.
Place of cooking					
In the house			Ref.		Ref.
In separate building			0.84 (0.77–0.93) **		0.90 (0.79–1.03)
Outdoors			0.83 (0.74–0.94) *		0.82 (0.69–0.97) *
Type of cooking fuel					
Clean fuels			Ref.		Ref.
Solid fuels			2.21 (1.48–3.31) **		1.29 (0.78–2.15)
Toilet facility					
Improved			Ref.		Ref.
Unimproved			1.16 (1.01–1.34) *		1.20 (0.98–1.47)
Open defecation			1.41 (1.21–1.63) **		1.57 (1.27–1.92) **
Source of drinking water					
Improved			Ref.		Ref.
Unimproved			0.95 (0.86–1.05)		0.88 (0.77–1.01)
Household flooring					
Improved			Ref.		Ref.
Unimproved			1.28 (1.08–1.51) *		1.13 (0.89–1.44)
Time to reach a water source					
On premises			Ref.		Ref.
≤30 min			1.08 (0.91–1.29)		0.90 (0.69–1.16)
31–60 min			1.22 (1.02–1.47) *		1.02 (0.78–1.34)
>60 min			1.26 (1.04–1.53) *		1.05 (0.79–1.38)
*Community Level Factors*					
Residence					
Urban				Ref.	Ref.
Rural				2.34 (2.07–2.65) **	1.41 (1.10–1.82) *
Region					
Large centrals				0.83 (0.73–0.95) *	0.79 (0.63–0.98) *
Small peripherals				1.01 (0.88–1.16)	0.89 (0.71–1.12)
Metropolis				Ref.	Ref.
Ecological zone					
Tropical zone				0.82 (0.71–0.97) *	0.92 (0.71–1.19)
Subtropical zone				0.70 (0.62–0.80) **	0.81 (0.65–1.01)
Cool zone				Ref.	Ref.
EDHS					
2005				0.99 (0.90–1.10)	0.98 (0.90–1.11)
2011				0.93 (0.86–1.01)	0.82 (0.72–0.93) *
2016				Ref.	Ref.
Random effect					
Variance (SE)	0.1632 (0.0001)	0.1177 (0.0014)	0.1992 (0.0001)	0.1426 (0.0006)	0.1851 (0.0018)
ICC (%)	4.72	3.45	5.71	4.15	5.32
MOR	1.59	1.49	1.68	1.55	1.65
PCV (%)	Ref.	27.87	22.05	12.62	13.41
Model comparison					
LL	−10,915.60	−5699.8017	−8563.7301	−10,772.627	−4430.9835
Deviance	21,831.20	11,399.6	17,127.46	21,545.254	8861.96
AIC	21,835.2	11,459.6	17,157.46	21,563.25	8959.967
BIC	21,850.98	11,677.97	17,272.55	21,634.25	9305.986

** *p*-value < 0.001, * *p*-value < 0.05; null model, containing no explanatory variable. ^a^: Model I adjusted for individual-level characteristics; ^b^: model II adjusted for household-level characteristics; ^c^: model III adjusted for community-level characteristics; and ^d^: model IV adjusted for individual-, household-, and community-level characteristics; SE: standard error; ICC: intraclass correction coefficient; MOR: median odds ratio; PCV: proportional change in variance; AIC: Akaike’s information criterion; BIC: Bayesian information criteria; LL: log-likelihood.

## Data Availability

The datasets analysed during the current study are publicly available on the Measure DHS website https://dhsprogram.com/data/dataset_admin/login_main.cfm (accessed on 28 March 2022), after formal online registration and submission of the project title and detailed project description.

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
