# Peer review of "Coexistence of Anaemia and Stunting among Children Aged 6–59 Months in Ethiopia: Findings from the Nationally Representative Cross-Sectional Study"

_ijerph, 2023, doi:10.3390/ijerph20136251_

Round 1

Reviewer 1 Report

Thank you for a high quality manuscript, please find the attached feedback document. 

Author Response

Reviewer 1

Thank you for the constructive comments, suggestions, and advice about how to improve the current manuscript. We have carefully considered the reviewers’ comments and revised the manuscript accordingly.

Reviewer 1

Dear authors

Thank you for submitting this manuscript titled Coexistence of anemia and stunting among children aged 6-59 months in Ethiopia: Findings from a nationally representative cross-sectional study to the International Journal of Environmental Research and Public Health.

The strength of this manuscript is to investigate the coexistence of anemia and stunting in children 6-59 months together with identifying related risk factors by using a national dataset in Ethiopia. The manuscript was well-written with good supporting literature. However, in the discussion, I would appreciate some relevant applications/recommendations of the findings, for example, what would be possible solutions in helping with disadvantaged children in Ethiopia? To reflect on the comments in the introduction that there is already nutritional program in place, what can be done to change the current high rate of this comorbidity.

I have included some detailed comments below:

Abstract

Clear information is presented in this abstract, however, I would like to know a bit about the characteristics of the study cohort.

Response: Thank you for your comment. We elaborate a little bit about the surveys I the method and result section.

Lines 22-24 did not give the results or odds of CAS.

Response:  we do not catch your point in lines 22-24. But we include a statement about the reporting of AOR with 95%CI.

Lines 28-29 what about female children? As the conclusion seems to indicate female children have a high risk of having CAS.

Response: We used female children as the reference category and being a male child [AOR: 0.81, 95% CI: (0.73-0.91)] was associated with lower odds of CAS. We also reported this in the abstract result section. We also revised the conclusion accordingly.

Line 33 the results only showed the higher odds of CAS for those who live in rural areas and born to mothers who had no education. This does not necessarily translate to participants from low socioeconomic disadvantage groups. Also, household wealth index was not a statistically significant risk factor for experiencing CAS.

Response:

Thank you for your advice. We remove the statement that stated “low socioeconomic disadvantaged group” from the conclusion section of the abstract.

Introduction

Lines 66-67 fits better to the second paragraph of the introduction.

Response: thank you for your suggestion. We merge it with the second paragraph of the introduction.

Response:

Line 69 would be useful to provide the data for the prevalence and the age of the young children, to show the current up-to-date data is important.

Response: As per your wise advice we rewrite lines 66-73. Please see the revised manuscript.    

Line 63 and line 72-73 both refer to a high prevalence of anemia and stunting feels a little repetition, could combine the discussion? Or maybe, the authors tried to demonstrate the separated prevalence of anemia and stunting, and a high prevalence of combined conditions, if this is the case, this is not clear in the current expression.

Response: As per your advice we have rewritten lines 66-73. Please see the revised manuscript.

Materials and methods

What was the total number of children under 5 years in the original data set? How was Maternal

BMI measured and classified? How was maternal anemia defined?

Response: We have provided a brief description on the number of children involve din each wave and how maternal BMI and maternal anemia were define

According to the EDHS-2005, a total of 4,138 children aged 6-59 months were measured for anthropometric data and Haemoglobin (n=4,138) [Ref: Central Statistical Agency [Ethiopia] and ORC Macro. 2006. Ethiopia Demographic and Health Survey 2005. Addis Ababa, Ethiopia and Calverton, Maryland, USA: Central Statistical Agency and ORC Macro. Page 157]

In the EDHS-2011, 9,157 children aged 6-59 months were anthropometric data and Haemoglobin data collected (n= 9,157) [Ref: Central Statistical Agency [Ethiopia] and ICF International. 2012. Ethiopia Demographic and Health Survey 2011. Addis Ababa, Ethiopia and Calverton, Maryland, USA: Central Statistical Agency and ICF International. Page 173]

In the EDHS-2016, anthropometric data and Hemoglobin was measured for children 9,267 aged 6-59 months (n=9,267). The testing was successfully completed for 88% of eligible children. [Ref: Central Statistical Agency (CSA) [Ethiopia] and ICF. 2016. Ethiopia Demographic and Health Survey 2016. Addis Ababa, Ethiopia, and Rockville, Maryland, USA: CSA and ICF page 195 and 210]. Please also visit supplementary file 1.

Results

Table 1

When was the maternal BMI self-reported/measured? Pre-pregnancy or postpartum?Maternal BMI classified as normal weight/overweight/obese in the discussion, perhaps provide classification when reporting the results.

Response: The EDHS collected anthropometric data on height and weight for women age 15-49 who are were not pregnant. These data were used to calculate maternal Body Mass Index (BMI). The BMI is was calculated by dividing weight in kilograms by height in metres squared (kg/m2). Maternal BMI was classified as underweight (<18.5kg/m2), normal (18.5 to < 24.9 kg/m2) or overweight/obesity ≥ 25.0 kg/m2). In the revised manuscript we indicate how maternal BMI was measured and classified.

What does stand for?

I understand the objective was to identify the prevalence of CAS, however, it would be interesting to know the prevalence of anemia, and the prevalence of stunting in this current national cohort.

Response: thank you for your wise advice. We have included the prevalence of anemia and stunting in the result section.

Discussion

Line 220 being a female child?

Response: We revised it accordingly please see the revised manuscript.

Line 223 and mothers with primary and secondary education?

Response: We have corrected accordingly.

Line 229 which group was compared to be higher?

Response: We have revised accordingly.

Line 239 and Line 286 LIMICs or LMICs?

Response: Thank you.  We have  corrected accordingly.

Line 259 what about mothers with BMIs > 25? 0.60 (0.45-0.81)? even lower than those with normal BMI at 0.82 (0.73-0.92)? Any explanation?.

Response: we have include these aspects in the conclusion section. However, because of the nature of the cross-sectional study design, we are not able to explain the reasons why these outcomes.

Line 275 cooking outdoors and open defecation indicated high risk of CAS , however, these were not discussed. Were these practices related to cultural preferences?

Response: The odds of having CAS among children from households that cooked outdoors [AOR: 0.82, 95% CI: (0.69-0.97)] was lower than children from households that cooked in the house. We have discussed these issues in the revised manuscript please see line 253 and 257.

It was interesting that a relatively small sample size was included in 2005, was it due to low birth rate during that period? There was also a significant finding that 2011 CAS differs from 2016, was any situation changed? Or initiatives implemented?

Response: you are right. In the EDHS 2005 a relatively lower number of children aged 6-59 months were included. This may be due to sampling issues as this was the first time they captured data for Haemoglobin testing.

  1. In the 2005 EDHS a representative sample of approximately 14,500 households from 540 clusters was selected. The sample was selected in two stages. In the first stage, 540 clusters (145 urban and 395 rural) were selected from the list of enumeration areas (EA) from the 1994 Population and Housing Census sample frame.
  2. The 2011 EDHS sample was selected using a stratified, two-stage cluster design, and EAs were the sampling units for the first stage. The sample included 624 EAs, 187 in urban areas and 437 in rural areas.
  3. The 2016 EDHS sample in the first stage, a total of 645 EAs (202 in urban areas and 443 in rural areas) were selected with probability proportional to EA size (based on the 2007 PHC) and with independent selection in each sampling stratum.

The odds of having CAS among children included from EDHS-2011 were lower compared with children from EDHS-2016[AOR: 0.82, 95% CI: (0.72-0.93)].  This may have been caused by the current increase in the prevalence of anemia. Between 2005 and 2016, the prevalence of anemia among Ethiopian children declined from 54% to 44% from 2005 to 2011 but increased to 57% in 2016.

Reviewer 2 Report

Thank you for analysing the EDHS data for this interesting study on coexistence of anaemia and stunting. The manuscript reads well and can provide useful information for its readers. If more recent data with relevant variables are already available, then I would suggest that the authors update their analysis using those data. Below, I share some other specific comments and suggestions.

Line 65: Please clarify which age group of children you refer to.

Line 89: It seems that surveys later than 2016 is now available, like 2021 survey. A revised analysis including more updated data would be reasonable. If, for some reason, that was not possible, please clarify. Also, here it needs clarification that data from several EDHS were combined here. 

Line 98: The total sample being 21,172 seems low. Please add the information on missingness of variables (overall, per EDHS round) as supplementary information. These could affect how representative the analysed sample was. This needs to be addressed doing some missing data imputation or mentioned as a limitation.

Line 112: It mentioned adjustment for altitude, but line 113 still mentioned the usual cut-off of 11g/dl. Please clarify or revise the sentence.

Line 125-126: It seems that the authors have combined the bottom two and top two wealth index groups, and use three wealth group categories which is not common practice. I think it will be good to keep the conventional grouping here, so people will find it easy to interpret and relate the findings with published findings reported elsewhere.

Table 1, birth interval category: It is not clear how the birth interval categories were decided, it seems quite arbitrary.

Table 1, size of child at birth: It seems that this heading should rather be 'perceived size of child at birth'.

Table 1, maternal BMI should be labelled as Thin (<8.5), normal (18.5-24.9), overweight or obese (>=25.0).

Line 169-170: Besides the overall prevalence, the specific EDHS related prevalence should be presented. Will be good to move up the EDHS specific prevalence estimate and mention them one followed by other.

Author Response

Reviewer 2

Thank you for the constructive comments, suggestions, and advice about how to improve the current manuscript. We have carefully considered the reviewers’ comments and revised the manuscript accordingly.

Reviewer 2

Comments and Suggestions for Authors

Thank you for analysing the EDHS data for this interesting study on coexistence of anaemia and stunting. The manuscript reads well and can provide useful information for its readers. If more recent data with relevant variables are already available, then I would suggest that the authors update their analysis using those data. Below, I share some other specific comments and suggestions. 

Line 65: Please clarify which age group of children you refer to.

Response: We revised this issue in the revised manuscript accordingly ( line 60-61).

Line 89: It seems that surveys later than 2016 is now available, like 2021 survey. A revised analysis including more updated data would be reasonable. If, for some reason, that was not possible, please clarify. Also, here it needs clarification that data from several EDHS were combined here. 

Response: thank you for your comment. The most current available data in Ethiopia were the EDHS 2016. We have clarified this in the revised manuscript.

Line 98: The total sample being 21,172 seems low. Please add the information on missingness of variables (overall, per EDHS round) as supplementary information. These could affect how representative the analysed sample was. This needs to be addressed doing some missing data imputation or mentioned as a limitation.

Response: A total of sample of 22,562 was included in the study. Haemoglobin measure was captured for children aged 6-59 months,  reducing the number of children in the sample. In addition, in some surveys such as EDHS -2016 the testing was completed for 88% of eligible children only. We have included the supplementary file 1 as per your suggestion.

In the EDHS-2005, anthropometric data and Haemoglobin measures were taken for children aged 6-59 months (n=4,138)[Ref: Central Statistical Agency [Ethiopia] and ORC Macro. 2006. Ethiopia Demographic and Health Survey 2005. Addis Ababa, Ethiopia and Calverton, Maryland, USA: Central Statistical Agency and ORC Macro. Page 157]

In the EDHS-2011: anthropometric data and Haemoglobin measures of  children aged 6-59 months were recorded (n= 9,157) [Ref: Central Statistical Agency [Ethiopia] and ICF International. 2012. Ethiopia Demographic and Health Survey 2011. Addis Ababa, Ethiopia and Calverton, Maryland, USA: Central Statistical Agency and ICF International. Page 173]

In the EDHS-2016: anthropometric data and Haemoglobin  measures of Children aged 6-59 months were undertaken  (n=9,267). [Ref: Central Statistical Agency (CSA) [Ethiopia] and ICF. 2016. Ethiopia Demographic and Health Survey 2016. Addis Ababa, Ethiopia, and Rockville, Maryland, USA: CSA and ICF page 195 and 210]

A total of 1390 observations were dropped due to height out of plausible limits, flagged cases, missing values for height or weight, and incomplete information on the outcome. In our analysis, only 2.4% were missing data (which is less than 10%), which did not affect statistical analysis. We understand that the analysis is likely to be biased when more than 10% of the data is missing. [Ref: Bennett DA. How can I deal with missing data in my study? Aust N Z J Public Health. 2001;25(5):464–9. https://doi.org/10.1111/j.1467-842X.2001.tb00294.x ]

Line 112: It mentioned adjustment for altitude, but line 113 still mentioned the usual cut-off of 11g/dl. Please clarify or revise the sentence.

Response: thank you for this observation.  We have revised accordingly.

Line 125-126: It seems that the authors have combined the bottom two and top two wealth index groups, and used three wealth group categories which are not common practice. I think it will be good to keep the conventional grouping here, so people will find it easy to interpret and relate the findings with published findings reported elsewhere.

Response: Thank you for this observation. For this analysis, we have re-coded the wealth index into three categories for adequate sampling in each category: 'poor' (poor and very poor), 'middle', and 'rich' (rich and very rich). In addition, in regression analysis, predictor variables having categories of five and above may not give reliable estimates. We acknowlde that other studies have followed a similar procedure [Ref: Mohammed SH, Larijani B, Esmaillzadeh A. Concurrent anemia and stunting in young children: prevalence, dietary and non-dietary associated factors. Nutrition journal. 2019;18(10):1-10.]

Table 1, birth interval category: It is not clear how the birth interval categories were decided, it seems quite arbitrary.

Response: Thank you for this observation. The birth interval categories were decided based on the World Health Organization (WHO) recommendations of at least 33 months of an inter-birth interval between two consecutive live births. Inter-birth interval of <33 months is considered a short birth interval, and between 36 and 59 months is considered the optimum birth interval. [Ref: World Health Organization. Report of a WHO Technical Consultation on Birth Spacing. Departemnt of Reproductive Health and Research. Geneva, Switzerland: WHO (2007).]

Table 1, size of child at birth: It seems that this heading should rather be 'perceived size of child at birth'.

Response: Thank you for this observation We have corrected this accordingly.

Table 1, maternal BMI should be labelled as Thin (<8.5), normal (18.5-24.9), overweight or obese (>=25.0).

Response: thank you for your suggestion. We  have included a footnote in table 1.

Line 169-170: Besides the overall prevalence, the specific EDHS related prevalence should be presented. Will be good to move up the EDHS specific prevalence estimate and mention them one followed by other.

Response: Thank you we have provided the survey-specific prevalence in the results section (please see Fig 1).

Reviewer 3 Report

The abstract provides a clear and concise overview of the research article, highlighting the prevalence and associated factors of coexisting anemia and stunting among children in Ethiopia.

The use of the Ethiopian Demographic and Health Survey (EDHS) data from 2005-2016 adds robustness to the study and allows for a large sample size.

The findings of a 24.4% prevalence of co-morbid anemia and stunting emphasize the significant public health problem faced by preschool-age children in Ethiopia.

The identification of various factors associated with co-morbid anemia and stunting, such as age, maternal characteristics, household practices, and socioeconomic status, provides valuable insights for designing targeted interventions.

The multilevel mixed-effects logistic regression model used in the analysis adds strength to the study, allowing for the consideration of individual and contextual factors.

The inclusion of odds ratios and confidence intervals provides a quantitative measure of the association between the identified factors and co-morbid anemia and stunting.

The recommendations for future interventions targeting female children, those perceived to be small at birth, anemic mothers, mothers with short stature, and those from low socioeconomic backgrounds are important for addressing the issue effectively.

It would be beneficial to provide some information on the limitations of the study, such as any potential biases or data collection challenges, to provide a more comprehensive understanding of the research.

It would be helpful to mention any implications of the findings for policy-making or the potential impact on existing interventions addressing stunting and anemia in Ethiopia.

Overall, the study contributes to the existing literature on co-morbid anemia and stunting in Ethiopia, and the findings have important implications for public health interventions and policy decisions.

nil

Author Response

Reviewer 3

Thank you for the constructive comments, suggestions, and advice about how to improve the current manuscript.

Reviewer 3

The abstract provides a clear and concise overview of the research article, highlighting the prevalence and associated factors of coexisting anemia and stunting among children in Ethiopia.

The use of the Ethiopian Demographic and Health Survey (EDHS) data from 2005-2016 adds robustness to the study and allows for a large sample size.

The findings of a 24.4% prevalence of co-morbid anemia and stunting emphasize the significant public health problem faced by preschool-age children in Ethiopia.

The identification of various factors associated with co-morbid anemia and stunting, such as age, maternal characteristics, household practices, and socioeconomic status, provides valuable insights for designing targeted interventions.

The multilevel mixed-effects logistic regression model used in the analysis adds strength to the study, allowing for the consideration of individual and contextual factors.

The inclusion of odds ratios and confidence intervals provides a quantitative measure of the association between the identified factors and co-morbid anemia and stunting.

The recommendations for future interventions targeting female children, those perceived to be small at birth, anemic mothers, mothers with short stature, and those from low socioeconomic backgrounds are important for addressing the issue effectively.

It would be beneficial to provide some information on the limitations of the study, such as any potential biases or data collection challenges, to provide a more comprehensive understanding of the research.

It would be helpful to mention any implications of the findings for policy-making or the potential impact on existing interventions addressing stunting and anemia in Ethiopia.

Overall, the study contributes to the existing literature on co-morbid anemia and stunting in Ethiopia, and the findings have important implications for public health interventions and policy decisions.

Response: Thank you for this suggestion. We have addressed the raised issues in the revised manuscript.